# Insulin-Loaded Soybean Trypsin Inhibitor-Chitosan Nanoparticles: Preparation, Characterization, and Protective Effect Evaluation

**DOI:** 10.3390/polym15122648

**Published:** 2023-06-11

**Authors:** Yihao Zhang, Ruijia Liu, Qixu Feng, He Li, You Li, Xinqi Liu

**Affiliations:** Key Laboratory of Geriatric Nutrition and Health, Ministry of Education, National Soybean Processing Industry Technology Innovation Center, Beijing Technology and Business University, Beijing 100048, China; zhangyihaobtbu@163.com (Y.Z.);

**Keywords:** soybean trypsin inhibitor, chitosan, insulin, complex coacervation, nanoparticles

## Abstract

The aim of this work was to prepare insulin-loaded nanoparticles using soybean trypsin inhibitor (STI) and chitosan (CS) as a potential coating. The nanoparticles were prepared by complex coacervation, and characterized for their particle size, polydispersity index (PDI), and encapsulation efficiency. In addition, the insulin release and enzymatic degradation of nanoparticles in simulated gastric fluid (SGF) and simulated intestinal fluid (SIF) were evaluated. The results showed that the optimal conditions for preparing insulin-loaded soybean trypsin inhibitor-chitosan (INs-STI-CS) nanoparticles were as follows: CS concentration of 2.0 mg/mL, STI concentration of 1.0 mg/mL, and pH 6.0. The INs-STI-CS nanoparticles prepared at this condition had a high insulin encapsulation efficiency of 85.07%, the particle diameter size was 350 ± 5 nm, and the PDI value was 0.13. The results of the in vitro evaluation of simulated gastrointestinal digestion showed that the prepared nanoparticles could improve the stability of insulin in the gastrointestinal tract. Compared with free insulin, the insulin loaded in INs-STI-CS nanoparticles was retained at 27.71% after 10 h of digestion in the intestinal tract, while free insulin was completely digested. These findings will provide a theoretical basis for improving the stability of oral insulin in the gastrointestinal tract.

## 1. Introduction

Diabetes mellitus is characterized by hyperglycemia levels in the body and is one of the most common chronic diseases all over the world [1]. The onset of diabetes can result from either the insufficient secretion of insulin by the islet β cells or impaired cellular responsiveness to insulin [2]. Diabetes mellitus is often accompanied by disturbances in the metabolism of carbohydrates, proteins, and fats [3]. In addition, diabetes mellitus can cause serious damage to the nervous and cardiovascular systems [2]. According to the International Diabetes Federation statistics, there are 537 million adults living with diabetes mellitus, which is expected to exceed 783 million by 2045 [1]. The number of deaths due to diabetes mellitus and its underlying diseases reached four million worldwide in 2017 and is climbing every year [4]. Diabetes mellitus is a serious threat to human health.

Insulin, a polypeptide hormone secreted by pancreatic cells in vertebrates, exerts hypoglycemic effects [1,5]. Since insulin was discovered by Canadians F.G. Banting and C.H. Best in 1921, it has been widely used in the clinical treatment of diabetes mellitus for the past 100 years [6]. Currently, in vitro insulin to lower postprandial hyperglycemia levels in diabetic patients is often administered by subcutaneous injections with excellent results [7]. However, the method is considered to be highly invasive [8]. Numerous reports indicate that subcutaneous insulin injections often result in the poor control of blood glucose levels in diabetic patients, the sudden invasion of insulin can lead to hypoglycemic symptoms in the organism, and frequent insulin injections are accompanied by the development of various complications, including pain, allergy, hyperinsulinemia, and lipodystrophy around the injection site [9]. Therefore, to effectively control diabetes, there is an urgent need to find a painless and convenient method of insulin administration. In recent years, considerable research efforts have been directed toward oral insulin delivery due to its ability to mimic the endogenous insulin pathway and provide better glucose homeostasis effects [5]. The oral route represents the most convenient and widely employed approach for insulin administration, demonstrating excellent patient compliance [5]. However, insulin is pH/protease sensitive, and orally administrated insulin is susceptible to degradation and loss of activity before reaching systemic circulation, which makes its oral bioavailability extremely low [10]. Effective delivery of insulin along the human GI tract remains a daunting task and challenging for food, pharmaceutical, and biological scientists [2]. The feasibility of using nanotechnology has gained substantial attention as a potential solution in the last few decades, including various natural and synthetic polymeric nanoparticles, liposomes, etc. [7].

Chitosan (CS), an abundant natural polysaccharide extracted from the shell of crustaceans or insects, represents prevalent polysaccharides in the world [11]. CS has some excellent properties such as biocompatibility, biodegradability, bio-adhesion, and pro-permeability [12,13]. Moreover, CS nanoparticles can be prepared in mild aqueous media, thus ensuring the stability of sensitive peptides/proteins during the encapsulation process, thereby preserving their structure and associated properties [10]. Yu-Hsin Lin et al. used CS and γ-glutamic acid to prepare nanoparticles for oral insulin delivery and showed that insulin-loaded NPs could effectively lower blood glucose levels in mice [14]. Piyasi Mukhopadhyay et al. cross-linked insulin with CS and found that the nanoparticles by crosslinking also could effectively lower the blood glucose levels in mice [15].

Soybean trypsin inhibitor (STI) is mainly a serine protease inhibitor, which can inhibit trypsin, chymotrypsin, and other proteases. According to amino acid composition, molecular weight, binding force with different proteases, and immunological reactivity, it can be divided into two main categories. One was the Bowman-Birk trypsin inhibitor (BBI), with 71 amino acids and a molecular weight of about 8 kDa, which had an inhibition of trypsin and chymotrypsin. The other type was the Kunitz trypsin inhibitor (KTI), with 181 amino acids and a molecular weight of about 20 kDa, which mainly had the properties of inhibiting trypsin [16,17,18]. STIs are stable at a wide range of pH and temperature, and showed resistance to pepsin degradation [19,20]. Additionally, they have anti-cancer, anti-inflammatory, anti-bacterial, anti-obesity, and hypoglycemic functional properties [21,22,23,24,25]. Studies have demonstrated that the concomitant administration of STI with teriparatide enhances the oral bioavailability of teriparatide [26]. The co-administration of insulin with a protease inhibitor and Ca^2+^ chelator results in the reduced hydrolysis of insulin by protease, leading to the improved oral bioavailability of insulin [27].

Although nanoparticles can effectively protect insulin from degradation in the gastrointestinal tract during the oral delivery of insulin through the wall effect of their shell, there remains a challenge concerning the potential degradation of insulin by trypsin during slow intestinal absorption, resulting in diminished insulin activity. Therefore, in this paper, we attempt to construct novel insulin-loaded nanoparticles using CS and STI by electrostatic complexation. The INs-STI-CS nanoparticles can use the dual action of the wall effect of its shell as well as the STI to inactivate the trypsin to protect insulin from protease hydrolysis to a greater extent and protect the integrity of insulin in the gastrointestinal environment in an attempt to improve the bioavailability of insulin in the organism.

## 2. Materials and Methods

### 2.1. Feedstocks and Reagents

CS (Mw = 100,000 Da, degree of deacetylation ≥ 80%) and insulin (≥27 USP units/mg) were purchased from Shanghai Macklin Biochemical Technology Co., Ltd. (Shanghai, China). Pepsin (1:3000) and trypsin (1000 BAEEu/mg) were bought from Shanghai Yuanye Biotechnology Co., Ltd. (Shanghai, China). All materials were of analytical grade and were chromatographically pure.

### 2.2. STI Preparation [28]

STI preparation was obtained according to previous procedures. The purity and the specific activity of STI were 71.11% and 1442.5 TIU/mg, respectively.

### 2.3. Preparation of Insulin-Loaded Soybean Trypsin Inhibitor-Chitosan Complex Coacervation [29,30]

Self-assembled nanoparticles were prepared by complex coacervation [15]. Briefly, CS was dissolved in 0.2% acetic acid solution to prepare 0.25, 0.5, 1.0, 2.0, and 4.0 mg/mL CS solutions, respectively. The pH was adjusted to 5.5 with 1 M NaOH. Insulin was dissolved in 0.01 M HCl and adjusted to pH 8.5 with 0.1 M Tris (hydroxymethyl) of the aminomethane solution to obtain 1.0 mg/mL of the insulin solution. STI solutions of 0, 0.5, 1.0, 1.5, and 2.0 mg/mL were prepared by dissolving STI in deionized water. A total of 2.5 mL of the insulin solution (1.0 mg/mL) was added to 5.0 mL of the STI solution (1.0 mg/mL) by magnetic stirring (C-MAG HS 7, IKA, Staufen, Germany) for 30 min, and then was slowly dropped into 5.0 mL of the CS solution (1.0 mg/mL) with a 1 mL syringe under the effect of magnetic stirring. Constant stirring was carried out for 30 min to complete the preparation of INs-STI-CS nanoparticles. The encapsulation efficiencies of insulin, particle size, and polydispersity index (PDI) were determined.

### 2.4. Determination of Insulin by RP-HPLC

Insulin content was determined by reversed-phase high-performance liquid chromatography (RP-HPLC) using the previously described method with appropriate modifications [31]. The conditions employed were as follows: The chromatographic column used was TC-C18 (250 mm × 4.6 mm, 5 µm). The mobile phase used was the aqueous solution with 0.1% trifluoroacetic acid (TFA) (A) and acetonitrile with 0.1% TFA (B). It was operated in gradient form, starting with 95:5 (A, B), and the run was ended with 40:60 (A, B) (a total of 30 min). The flow rate was kept constant at 1.0 mL/min and the injection volume was 20 µL. The temperature of the column was 40 °C and the detection wavelength was 280 nm. Standard insulin (31.25, 62.5, 125, 250, and 500 μg/mL) was determined using this method. Taking insulin mass concentration as a horizontal coordinate and peak area as a vertical coordinate, the standard curve was drawn (Y = 17.389X − 267.51; R = 0.9999), and the amount of insulin was calculated in the sample.

### 2.5. Calculation of Encapsulation Efficiency

The prepared insulin nanoparticle solution was placed in the ultrafiltration centrifuge tube (the intercepted relative molecular mass was 100 kDa, Millipore, Burlington, MA, USA) and centrifuged by centrifuge (Allegra X-30R, Beckman Coulter, Brea, CA, USA) for 30 min (4 °C, 4500 rpm) [32]. The amount of free insulin in the filtrate was determined using RP-HPLC. Record the peak area of insulin and calculate the concentration of free insulin according to the standard curve. The encapsulation efficiency value was obtained using the following formula:$$\mathrm{Encapsulation}\;\mathrm{Efficiency}=\frac{M_{0}-M_{1}}{M_{0}}\times 100\%$$
where M_0_ is the initial insulin amount/(mg), M_1_ is the amount of free insulin/(mg).

### 2.6. Effects of CS, STI, and pH on Insulin Encapsulation Efficiency and Particle Characteristics

#### 2.6.1. Optimization of CS Concentration

A total of 2.5 mL of the insulin solution (1.0 mg/mL) was added to 5.0 mL of the STI solution (1.0 mg/mL) by magnetic stirring for 30 min, and then was slowly dropped into 5.0 mL of the CS solution (0.25, 0.5, 1.0, 1.5, and 2.0 mg/mL) with a syringe under the effect of magnetic stirring. Constant stirring for 30 min completed the preparation of INs-STI-CS nanoparticles. The encapsulation efficiencies of insulin, particle size, and PDI were determined.

#### 2.6.2. Optimization of STI Concentration

A total of 2.5 mL of the insulin solution (1.0 mg/mL) was added to 5.0 mL of the STI solution (0, 0.5, 1.0, 1.5, and 2.0 mg/mL) by magnetic stirring for 30 min, and then was slowly dropped into 5.0 mL of the CS solution (2.0 mg/mL) with a syringe under the effect of magnetic stirring. Constant stirring was carried out for 30 min to complete the preparation of INs-STI-CS nanoparticles. The encapsulation efficiencies of insulin, particle size, and PDI were determined.

#### 2.6.3. Optimization of pH of the Complexation in the Mixing System

The optimized CS solution (2.0 mg/mL, pH 5.5) and STI solution (1.0 mg/mL, pH 6.5) from Section 2.6.1 and Section 2.6.2 were used to optimize the pH environment of the complexation in the mixing system. INs-STI-CS nanoparticles were prepared at pH values ranging from 4.5 to 6.5, by careful controlling of the pH with 1 M HCl and 1 M NaOH. The encapsulation efficiencies of insulin, particle size, and PDI were determined.

### 2.7. Characterization of INs-STI-CS Nanoparticles

The particle size and PDI were determined by the dynamic light scattering method, using a PCS8501 sample cell and nano-laser particle size analyzer. The morphological characteristics of nanoparticles were identified by scanning electron microscopy (SEM) and transmission electron microscopy (TEM).

#### 2.7.1. Particle Size and PDI Value

The samples were diluted (5-fold) in sodium acetate buffer solution to avoid multiple scattering. Zetasizer Nano ZS90 apparatus (Malvern Instruments, Worcestershire, UK) was used to determine the average particle size and PDI value of INs-STI-CS nanoparticles. The sample was measured at 25 °C for 10 times [28].

#### 2.7.2. Microscopic Morphological Characteristics

In order to observe the surface morphology of the samples, SEM tests were performed on insulin, STI, CS, and INs-STI-CS nanoparticles. The freeze-dried samples were sprayed with gold for 45 s using an Oxford Quorum SC7620 sputter coater, followed by a Zeiss Merlin Compact scanning electron microscope (Zeiss Merlin Compact, Jena, Germany) used to photograph the sample morphology with an accelerating voltage of 3 kV during morphology photography [28].

TEM was used to evaluate the morphological characteristics of INs-STI-CS nanoparticles. In this method, an insulin-loaded nanoparticle solution (10 μL) was moved onto a carbon-coated copper grid, and the sample was viewed and photographed with a Tecnai Spirit D1319 transmission electron microscope at an accelerated voltage of 80 kV [5].

### 2.8. Nanoparticles In Vitro Simulated Gastrointestinal Sustained Release Insulin [2]

Pepsin-free SGF (pH 1.2) and trypsin-free SIF (pH 6.8) were prepared according to the United States Pharmacopeia protocol. SGF without protease was obtained via dissolving 2 g of NaCl and 7 mL of concentrated HCl in 900 mL of deionized water, adjusting the pH to 1.2 ± 0.1 with 1 M HCl, and then adjusting the volume to 1000 mL with deionized water. SIF without protease was prepared in a similar way: 6.8 g of KH_2_PO_4_ was dissolved in 250 mL of deionized water, 77 mL of 0.2 M NaOH was diluted to 500 mL, both solutions were mixed, and then the pH was adjusted to 6.8 ± 0.1 with 1 M NaOH. Finally, the volume was adjusted to 1000 mL with deionized water to complete the trypsin-free SIF preparation.

To determine the release profile of insulin, 17.5 mg of INs-STI-CS nanoparticles (lyophilized product) was sequentially dispersed in 2.5 mL of pepsin-free SGF (pH 1.2) and trypsin-free SIF (pH 6.8) and gently stirred in a constant temperature incubator at 37 °C. Samples were taken at predetermined time points and placed in 100 kDa ultra-filtration centrifuge tubes (Millipore, Burlington, MA, USA) for ultrafiltration, centrifuged for 30 min (4 °C, 4500 rpm), and the filtrate was aspirated to determine the insulin release from nanoparticles by RP-HPLC analysis.

### 2.9. Structural Stability of Insulin Released from the Nanoparticles

The insulin-loaded nanoparticles were dissolved in pepsin-free SGF (pH 1.2) and trypsin-free SIF (pH 6.8) and gently stirred in a constant temperature incubator at 37 °C for 60 min and 240 min, respectively. The solutions were ultrafiltered in 100 kDa ultrafiltration tubes (Millipore, Burlington, MA, USA) and centrifuged for 30 min (4 °C, 4500 rpm). The filtrate was adjusted to pH 5.4 with 1 M NaOH and centrifuged for 10 min (4 °C, 12,000 rpm). The precipitate was redissolved in 0.01 M HCl and centrifuged for 10 min (4 °C, 12,000 rpm). The supernatant was freeze-dried and used for the subsequent assessment of insulin structural stability. The conformational stability of insulin was determined by circular dichroism (CD) (MOS-500, BioLogic, Paris, France) spectroscopy, and the tested insulin concentration was 500 μg/mL, scanned in the wavelength range of 190~250 nm [4].

### 2.10. In Vitro Gastrointestinal Digestion of INs-STI-CS Nanoparticles

#### 2.10.1. Simulated Gastric Digestion

To evaluate the protective effect of INs-STI-CS nanoparticles on insulin in the stomach [33], 2.5 mg of insulin and 17.5 mg of INs-STI-CS nanoparticles were dissolved in 2.5 mL of SGF (pH 1.2, pepsin content: 0.32 mg/mL), respectively. The solutions were incubated in a 37 °C constant temperature incubator. A total of 480 µL of the sample was taken at predetermined time points and the enzyme reaction was terminated using 120 µL of 0.1 M NaOH. The mixed solution was shaken continuously in a 37 °C constant temperature incubator for 2 h. The solution was filtered with a 0.22 µm filter membrane, and the contents of residual insulin in the filtrate were determined by RP-HPLC.

#### 2.10.2. Simulated Intestinal Digestion

The protective effect of INs-STI-CS nanoparticles on insulin in the intestine was evaluated by dissolving 2.5 mg of insulin and 17.5 mg of INs-STI-CS nanoparticles in 2.5 mL of SIF (pH 6.8, trypsin content: 10 mg/mL), respectively [33]. The solutions were incubated in a 37 °C constant temperature incubator. A total of 480 µL of the sample was taken at a predetermined time point and the enzyme reaction was terminated using 120 µL of 5% TFA. The mixed solution was shaken continuously in a 37 °C constant temperature incubator for 24 h. The solution was filtered with a 0.22 µm filter membrane, and the contents of residual insulin in the filtrate were determined by RP-HPLC.

### 2.11. Statistical Analysis

The experimental data were processed by using SPSS Statistics 19 software (IBM, Chicago, IL, USA). The experimental data were expressed as “X ± SD”. Significant differences between means were stated at *p* < 0.05. Origin 2021 was used for data visualization.

## 3. Results and Discussion

### 3.1. Effects of Different Factors on Insulin Encapsulation Efficiency, Particle Size, and PDI Value

Figure 1a shows the effect of micro-/nanoparticles formed by cross-linking CS with STI at different concentrations on insulin encapsulation efficiency, particle size, and PDI values. The results showed that the encapsulation efficiency of insulin gradually increased with the increase in CS concentration, and when the CS concentration was 4.0 mg/mL, the encapsulation efficiency of insulin by the prepared particle reached 100%. The dynamic light scattering results showed that the diameter of the formed particles decreased and then increased (1100 ± 52 nm → 261 ± 6 nm → 308 ± 6 nm) with the increasing CS concentration. The particle size was the smallest (216 ± 6 nm) at a CS concentration of 2.0 mg/mL. The PDI value also showed the same trend as the particle size. The PDI value decreased and then increased with the increase in CS concentration, and the PDI value was 0.105 for a CS concentration of 2.0 mg/mL. This phenomenon may be attributed to the number of charges and the number of molecules carried by CS. After ionic gelation of the low concentration of CS with STI, the surface electrostatic charge of nanoparticles was low and the structure was loose and prone to aggregation, resulting in relatively low insulin encapsulation efficiency, a large particle diameter, and poor stability (the PDI values are relatively large). With the increase in CS concentration, the number of positive charges of the system increased, the surface potential energy of the formed composite nanoparticles increased, the particles repelled each other, their particle size decreased, the encapsulation efficiency increased, and the stability became better. When the CS concentration rose to a certain value, the particle size became larger and the stability decreased, probably because of the excess CS molecules, which did not form micro- and nanoparticles through the ionic gel; the linear CS increased the average particle size, while the system PDI value increased under the interference of CS, showing a decrease in particle stability. Figure 1b shows the effects of micro-/nanoparticles formed by cross-linking STI with CS at different concentrations on insulin encapsulation efficiency, particle size, and PDI values. At constant CS concentration (2.0 mg/mL), the encapsulation efficiency of insulin gradually increased with the increase in STI concentration. When the STI concentration was 1.0 mg/mL, the encapsulation efficiency of insulin by nanoparticles formed by complex coacervates reached 88.09%, and there was almost no change in the encapsulation efficiency when the STI concentration continued to increase. The dynamic light scattering results showed that the particles formed at the STI concentration of 1.5 mg/mL were less than 305 ± 6 nm in diameter and the PDI values were less than 0.03. When the STI concentration was 2.0 mg/mL, the particle diameter was 4112 ± 679 nm and the PDI value was 1.0, which made the solution unstable and prone to aggregation and precipitation. Therefore, STI at a concentration of 1.0 mg/mL was chosen as the subject of study in the subsequent experimental work. Figure 1c shows the effect of micro-nanoparticles formed under different pH conditions on insulin encapsulation efficiency, particle size, and PDI values. With the increase in pH value, the insulin encapsulation efficiency gradually increased (from 35.33% to 89.81%). The maximum insulin encapsulation efficiency was 89.81% at pH 6.5. However, the particle size of the complex coacervates formed at this time was larger, and the particle diameter was 4443 ± 916 nm; the PDI value was larger at 0.76. This indicated that the solution system was unstable at this time, and aggregation and precipitation could easily occur. At pH 6.0, the insulin encapsulation efficiency of complex coacervates was 85.07%. However, the particle size was 350 ± 5 nm and the PDI value was 0.13. Therefore, pH 6.0 was selected as the optimal condition for the preparation of INs-STI-CS nanoparticles. In summary, the optimal conditions for the preparation of INs-STI-CS nanoparticles were: a CS concentration of 2.0 mg/mL, STI concentration of 1.0 mg/mL, and pH of 6.0. The INs-STI-CS nanoparticles prepared at this condition had an insulin encapsulation efficiency of 85.07%, the particle diameter was 350 ± 5 nm, and the PDI value was 0.13. Compared with the polyelectrolyte complexes particles reported in previous studies, the size of the particles obtained in this study was relatively large. However, the grade was still in the nanometer-size range. Moreover, the nanoparticles obtained in this study demonstrated a considerably improved insulin encapsulation efficiency, exhibiting an increase of 29.87% [9].

### 3.2. Microstructure of Complex Coacervates Nanoparticles

#### 3.2.1. Scanning Electron Microscopy (SEM)

The SEM presentations of insulin, STI, CS, and INs-STI-CS nanoparticles were performed to observe the morphological characteristics of the surface of the lyophilized powder (Figure 2). As shown in Figure 2A (A_1_ = 500×, A_2_ = 5000×), the lyophilized insulin showed a loose morphological distribution with particles in the form of a cubic crystal structure, and the network structure could not be formed between the particles. The results are consistent with those observed by Mayyas [34]. Figure 2B (B_1_ = 500×, B_2_ = 5000×) shows the morphological structure of STI lyophilized powder. It can be seen that the STI lyophilized powder showed a lamellar structure, which was observed under SEM high magnification; the lamellar structure was formed by the aggregation of numerous tiny particles. Figure 2C (C_1_ = 500×, C_2_ = 5000×) shows the microstructure of CS lyophilized powder, and the results show that the CS powder displayed a filamentous structure with a smooth surface of its microstructure. Figure 2D (D_1_ = 500×, D_2_ = 5000×) shows the images of INs-STI-CS nanoparticles. The results show that the INs-STI-CS nanoparticles have a homogeneous honeycomb-type network structure, and the polymer freeze-dried powder structure varies in size, but all consist of homogeneous, smooth, spherical particle aggregated [8]. It is well known that the freeze-drying process generates freezing and dehydration stresses that can destabilize the colloidal suspension of nanoparticles, leading to the irreversible aggregation or fusion of nanoparticles [35].

#### 3.2.2. Transmission Electron Microscopy (TEM)

The morphological structure of INs-STI-CS nanoparticles was observed by transmission electron microscopy (TEM). As shown in Figure 3, the TEM image shows that the structure of INs-STI-CS nanoparticles was spherical or nearly spherical with sizes ranging from 100 to 200 nm, which slightly deviated from the dynamic light scattering measurements (the dynamic light scattering assay showed that the nanoparticle size was at 350 nm). The nanoparticle’s diameter observed under TEM was smaller in size than those obtained by dynamic light scattering detection—such results are consistent with those described by Pratyusa Sahoo et al. [4,36,37]. This may be because the nanoparticles form relatively small particle structures due to dehydration and aggregation when the sample is moved to the carbon-coated copper grid to dry, or because the less coalesced polysaccharide layer at the edges of the nanoparticles does not show up under TEM or does not show up clearly [5]. The structure of the individual particles showed an irregular viscous structure on the surface and a possible “drawing” phenomenon between the particles, which was similar to the results observed by SEM. The surface structure of the INs-STI-CS nanoparticles is smooth and sticky, which may be the result of CS covering the STI with insulin. The above results suggest that the INs-STI-CS nanocomposite cohesion may be composed of a core-shell structure.

### 3.3. In Vitro Simulated Gastrointestinal Slow Release

Figure 4A shows that the insulin release from INs-STI-CS nanoparticles increases gradually with time when incubated in SGF (pH 1.2). At the initial 10th min, the INs-STI-CS nanoparticles presented a 43.17% insulin release. After 90 min of simulated gastric juice treatment, 96.33% of insulin was released from the INs-STI-CS nanoparticles. The rapid release of insulin from nanoparticles at pH 1.2 and its almost complete release within 90 min may be attributed to the dissolution of CS at a low pH and loss of interaction with STI and insulin [1]. Previous studies have shown that CS nanoparticles are pH-responsive and release the encapsulated material rapidly at lower pH conditions [10]. In order to overcome the rapid release of encapsulated substances from CS-based carrier nanoparticles under gastric environment conditions, researchers have modified CS, such as glycosylation reaction, grafting reaction, etc., so that CS can better protect the encapsulated substances under gastric environment conditions.

Figure 4B shows that insulin release from INs-STI-CS nanoparticles increases slowly with time when incubated in SIF (pH 6.8). This result shows that INs-STI-CS do not release insulin abruptly in SIF (pH 6.8) and insulin release exhibits a continuous slow release. The insulin release from the INs-STI-CS nanoparticles was 25.52% at the 10th min of initial incubation, while the insulin release from the INs-STI-CS nanoparticles was 46.81% after 120 min, which indicates that the INs-STI-CS is structurally stable in SIF (pH 6.8) and the nanoparticle-encapsulated substance exhibits a regular and slow release. This phenomenon may be attributed to the fact that the isoelectric point of CS is at pH 6.4 and CS is difficult to melt or insoluble under neutral as well as alkaline conditions. Insulin and STI are soluble at pH 6.8, so the core-shell nanostructures formed by cross-linking insulin and STI with CS may dissolve over time, and the encapsulated substances will be released slowly.

### 3.4. Stability of Insulin Extracted from Nanoparticles

The primary and secondary structures of protein and peptide molecules directly affect the functional properties of proteins and peptides. Once the primary and secondary structures of proteins and polypeptides are altered, their original functions are largely lost. Insulin belongs to the class of protein peptides, a kind of hormone compound, whose primary and secondary structures directly affect its physiological functional properties [38]. This paper evaluates the primary and secondary structures of insulin released from INs-STI-CS, which is a guide to measure whether there is a loss of activity. The results of high-performance liquid chromatography showed that the retention time of insulin released from nanoparticles did not change, indicating that its insulin molecular weight size did not change, which means that the primary structure of insulin released from nanoparticles did not change [39]. It is also interesting to probe the secondary structure of insulin released from nanoparticles since the formation process of insulin-loaded nanoparticles is driven by electrostatic and hydrophobic interactions. Circular dichroism (CD) spectroscopy is a technique that is widely used for protein structure studies [39,40]. Figure 5 shows that the CD spectra of insulin released from INs-STI-CS nanoparticles in simulated gastric fluid (INs-SGF) and insulin released from INs-STI-CS nanoparticles in simulated intestinal fluid (INs-SIF) are largely consistent with those of standard free insulin (INs), indicating that the secondary structure of insulin released from nanoparticles is not altered. In conclusion, INs-STI-CS nanoparticles prepared by complex coacervations do not change the primary and secondary structures of insulin. The structure of insulin slowly released from nanoparticles does not differ from that of free insulin. Therefore, it is presumed that insulin released from the nanoparticles has no loss of functional properties.

### 3.5. In Vitro Simulation of Gastrointestinal Digestion

Pepsin and trypsin were used as model proteases to digest the nanoparticles. As shown in Figure 6A, free insulin was rapidly degraded in SGF (pH 1.2), with 34.07% free insulin remaining after 10 min; 1.17% free insulin remaining after 30 min; and after 60 min, the peak of insulin was no longer detected. The results indicated that free insulin was highly susceptible to degradation by pepsin, which was consistent with those reported in the literature [41]. In contrast, the insulin encapsulated in INs-STI-CS nanoparticles remained at 84.64% after 10 min and at 75.53% after 120 min. The results showed that the INs-STI-CS nanoparticles could well resist the degradation of pepsin during two hours of simulated gastric digestion, which allowed for the insulin in INs-STI-CS to retain its structural integrity for a longer period of time in the simulated gastric juice. Compared with free insulin, the nanoparticles showed an excellent protective effect.

As shown in Figure 6B, in SIF (pH 6.8), free insulin was slowly degraded with time, and after 2.5 h of trypsin treatment, more than half of the insulin was degraded and its retention rate was 46.84%, while after 10 h of simulated intestinal fluid treatment, free insulin was basically undetectable and the insulin retention rate was 0. Additionally, for the insulin encapsulated in INs-STI-CS nanoparticles, the insulin residual was 83.16% after 10 min, and the insulin retention in the nanoparticles was still above 50% after 4 h of trypsin treatment, while the insulin retention in the INs-STI-CS nanoparticles was 27.71% after 10 h of simulated intestinal fluid treatment. The results indicated that the INs-STI-CS could improve the stability of insulin in the simulated intestinal fluid. In conclusion, INs-STI-CS nanoparticles have excellent insulin protection in simulated gastrointestinal fluid, and it is speculated that INs-STI-CS nanoparticles have the potential ability to improve the bioavailability of oral insulin.

## 4. Conclusions

In the present study, we produced uniform nanoparticles by the polyelectrolyte complexation of STI, CS, and insulin. The STI concentration, CS concentration, and pH of the reaction system were optimized to form insulin-loaded nanoparticles with small size, excellent stability, and high encapsulation efficiency. In addition, insulin-loaded nanoparticles were characterized, and their protection was evaluated through simulating gastrointestinal digestion. The results showed that the diameter of INs-STI-CS nanoparticles was 350 ± 5 nm and the insulin encapsulation efficiency was 85.07% when the CS concentration was 2.0 mg/mL, STI concentration was 1.0 mg/mL, and pH was 6.0. The results of TEM and SEM showed that the structure of INs-STI-CS nanoparticles was spherical or nearly spherical, the size of which was between 100 and 200 nm, and the “drawing” phenomenon existed between particles. The results of the in vitro evaluation of simulated gastrointestinal digestion showed that the prepared nanoparticles could improve the stability of insulin in the gastrointestinal tract. Compared with free insulin, the insulin loaded in INs-STI-CS nanoparticles was retained at 27.71% after 10 h of digestion in the intestinal tract, while free insulin was completely digested. However, further studies are needed to determine whether, or how much, STI provides protection in protecting insulin stability.

## Figures and Tables

**Figure 1 polymers-15-02648-f001:**
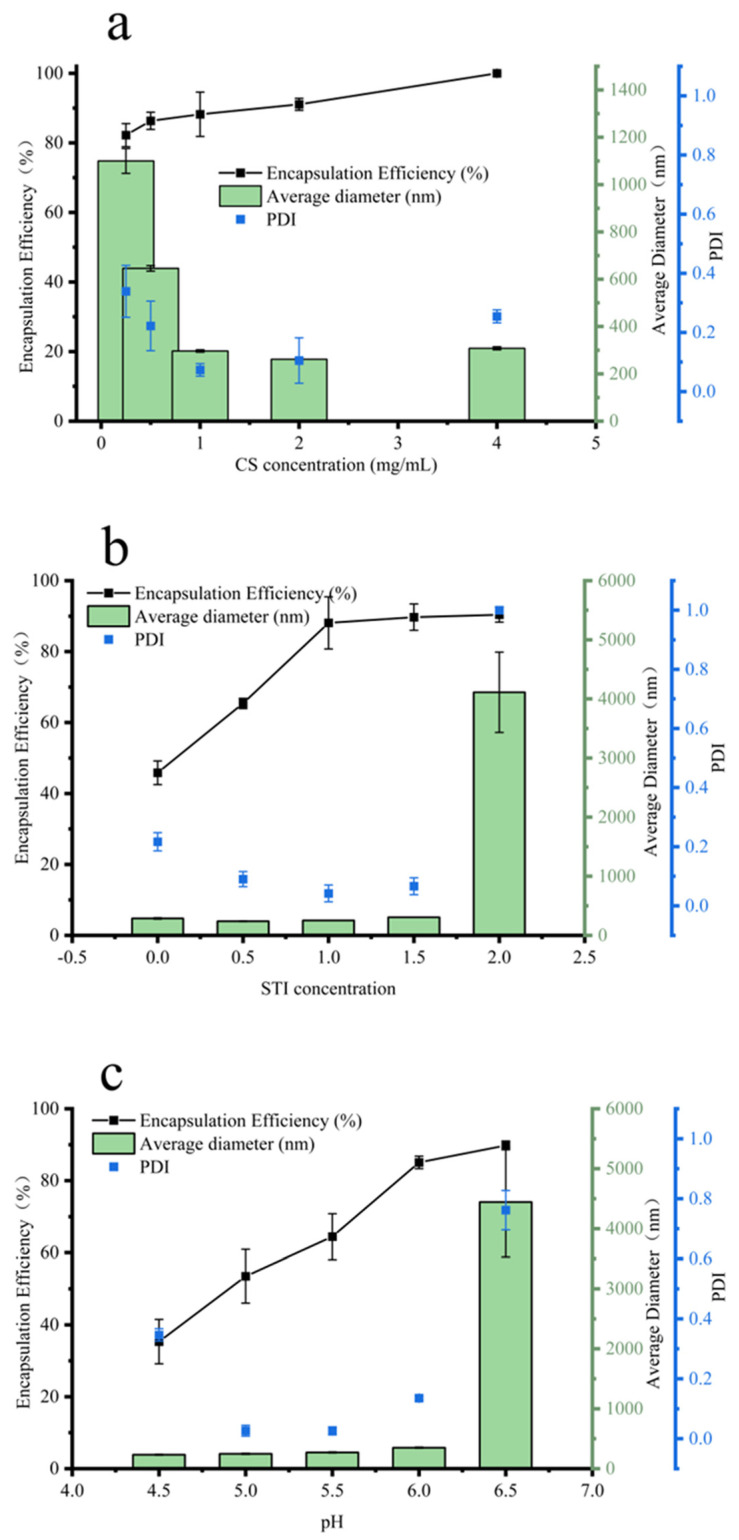
Optimization of parameters for generating INs-STI-CS nanoparticles. (**a**) Effect of CS concentration on the average size and encapsulation efficiency of the insulin; (**b**) effect of STI concentration on the average size and encapsulation efficiency of the insulin; (**c**) effect of pH on the average size and encapsulation efficiency of the insulin.

**Figure 2 polymers-15-02648-f002:**
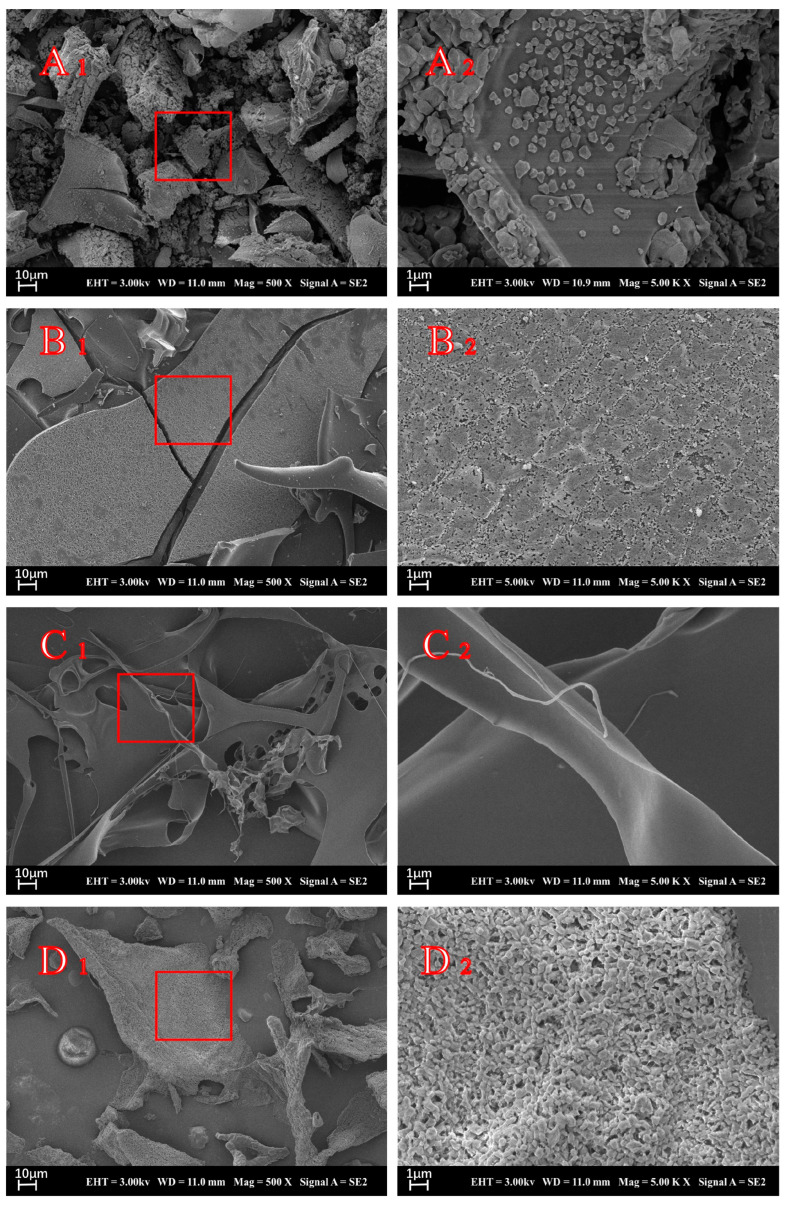
SEM micrographs of insulin (**A**), STI (**B**), CS (**C**), and INs-STI-CS (**D**).

**Figure 3 polymers-15-02648-f003:**
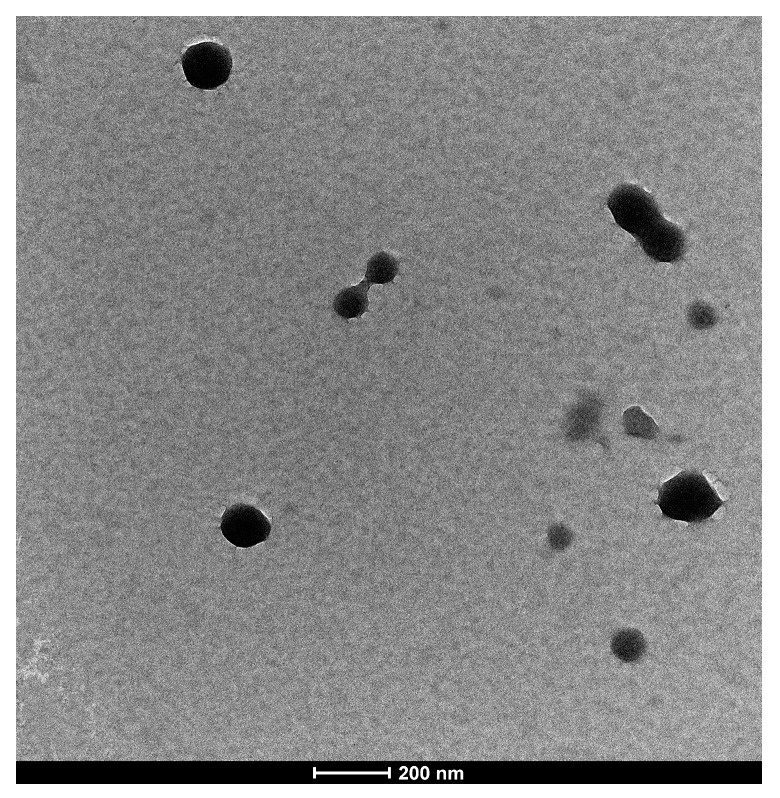
TEM micrographs of INs-STI-CS nanoparticles.

**Figure 4 polymers-15-02648-f004:**
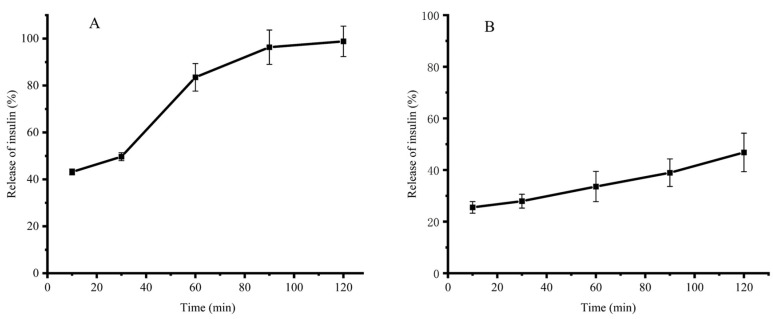
The release profile of insulin in the SGF condition (pH 1.2) (**A**) and SIF condition (pH 6.8) (**B**).

**Figure 5 polymers-15-02648-f005:**
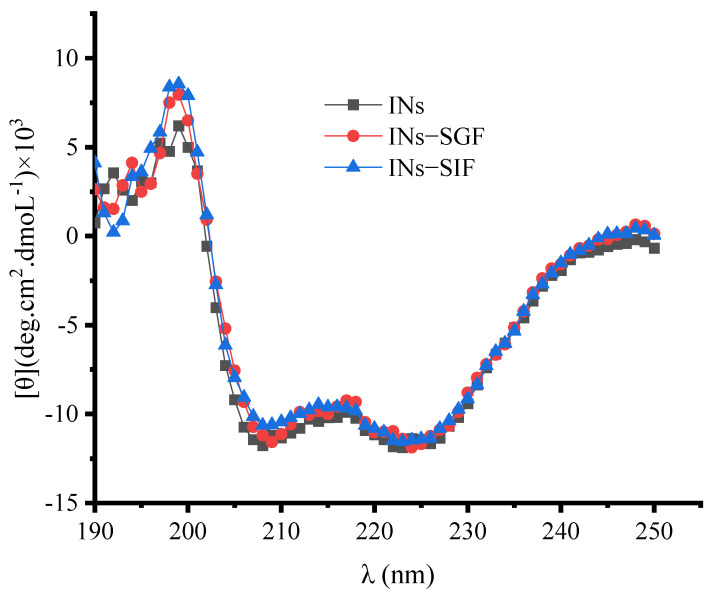
The circular dichroism spectra of insulin in different matrices.

**Figure 6 polymers-15-02648-f006:**
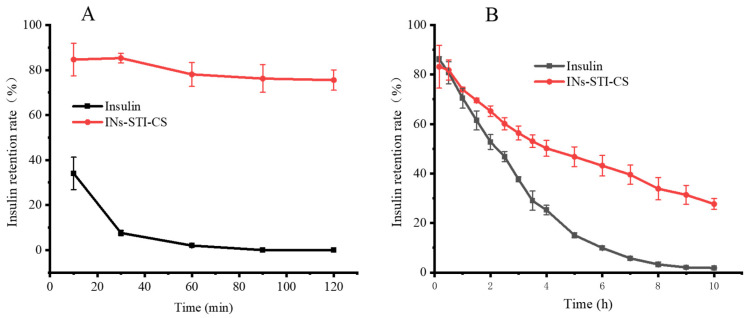
The retention rate profile of insulin and INs-STI-CS nanoparticles in SGF (pH 1.2) (**A**) and SIF (pH 6.8) (**B**).

## Data Availability

The data presented in this study are available on request from the corresponding author.

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
