# Peer review of "Insulin-Loaded Soybean Trypsin Inhibitor-Chitosan Nanoparticles: Preparation, Characterization, and Protective Effect Evaluation"

_polymers, 2023, doi:10.3390/polym15122648_

Round 1
Reviewer 1 Report
The manuscript "Insulin loaded soybean trypsin inhibitor-chitosan nanoparticles: preparation, characterization, and protective effect evaluation" is devoted to the preparation of nanoparticles for oral insulin delivery consisting from chitosan, trypsin inhibitor and insulin. The problem of finding ways to oral insulin administration is urgent, but still challenging and thus, the topic of the study is important.
1. However, the experimental data do not provide enough evidence of nanoparticle synthesis. First, in SEM images (Figure 2D1 and 2D2) there are no separate nanoparticles. It looks like there are a lot of large objects of variable sizes and forms, which resemble films with a structured surface, but not separate nanoparticles or nanoparticle agglomerates. It seems that insulin and trypsin inhibitor precipitated on chitosan polymeric network and formed a film with a structured surface similarly to that in paper (doi:10.1016/j.foodhyd.2008.02.009, Fig. 7) or even in a previous paper by the authors themselves (https://doi.org/10.3390/polym15071594), where they visualized the complex between chitosan and soybean trypsin inhibitor. Moreover, the form of these convex structures is not "spherical", but irregular. Second, according to the procedure of nanoparticle synthesis described in Materials and methods section, the mixture of insulin and trypsin inhibitor was dropped to the stirred chitosan solution from a syringe. It cannot be deduced what stabilized the surface of the forming individual nanoparticles, because chitosan is still soluble at pH 4,5-6. Could the authors please provide the SEM images of not lyophilized nanoparticles analogous to those that were used for TEM imaging?
2. Lines 70-79. This block of text is identical to a block in the paper (https://doi.org/10.3390/polym15071594) published previously by the same authors. Some sentences from the previous text block are also identical to sentences from the same paper. I do not know if self-plagiarism is acceptable by MDPI rules.
3. Some important details of the experimental procedure are missing, such as what type of a syringe was used, the model of the centrifuge, etc. Lines 150-151: "INs-STI-CS nanoparticles were prepared at the pH values ranging from 4.5 to 6.5, by careful controlling of the pH" – Please indicate clearly the pH of every solution (INs, STI, CS) used.
4. This study is not the first one where chitosan was used for oral insulin delivery. Other studies devoted to using chitosan for oral insulin delivery should be cited in the Introduction.
The English grammar should be checked because some sentences are confusing. Some examples:
Line 61: "...and is the second most widely spread in the world." – second most widely spread what? It seems like a word is missing between "spread" and "in the world"
Lines 67-69: " In addition, CS nanoparticles can be prepared in mild aqueous media, thus ensuring the stability of sensitive peptides/proteins during encapsulation of their structure, and properties"
Lines 80-82: "Although nanoparticles can largely protect insulin from degradation in the gastrointestinal tract during oral delivery of insulin by using the wall effect of its shell, insulin released slowly during intestinal absorption is likely to be degraded by trypsin because it is too late to be absorbed, thus losing its activity." – The sentence is not clear.
Lines 85: "The INs-STI-CS nanoparticles can use the dual action of the wall effect of its shell..." - What is "wall effect of its shell"?
Lines 278-280: In addition, the nanoparticles obtained in this study, which had a relatively excellent insulin encapsulation efficiency, increased by 29.87%[9] - The sentence is not clear.
These are only some examples, so the whole text should be carefully revised.
Author Response
please see the attachment, thank you!

Reviewer 2 Report
The manuscript describes the development of insulin-loaded nanoparticles for oral administration. Chitosan was used as the main polymeric coating material fo rthe nanoparticles, while soybean trypsin inhibitor was used to provide protection against enzymatic degradation in the GIT. The overall level of the manuscript is very good. However, some conclusions don't have enough data support. Also, the quality of the English language needs to be improved. The manuscript may need some few experiments (or one experiment) at least to be eligible for publication.
Recommendation: The manuscript can be published in Polymers after major revision.
Remarks:
1- Line 165: How were the nanoparticles lyophilized? This needs to be described.
2- When the nanoparticles were lyophilized, did the authors measure the particle size and zeta potential after lyophilization? Nanoparticles are known to massively aggregate upon lyophilization without the use of cryoprotectant (e.g., sugar).
3- Section 2.11. Limits of statistical significance need to be clarified.
4- Major: typically, LC/MS or LC/MS/MS are used to confirm that the structural integrity of proteins (e.g. insulin) is maintained. Studies showed that acidic conditions may result in the formation of desamido insulin, which may not differ in retention time in RP-HPLC (UV) compared with intact insulin (M.A.Ibrahim et al., Journal of Controlled Release, 106 (2005) 241–252). It is strongly recommended that the authors would use a more solid approach to confirm that the integrity of insulin is maintained in acidic/basic conditions. RP-HPLC (UV) may indicate major change/degradation of insulin structure, but not minor changes, which may still render insulin ineffective. Literature citations are needed to support the experimental approach that the authors used, otherwise, LC/MS may be unavoided.
1- Line 22: Please correct the expression 'retained of 27.71%'.
2- Lines 31-32 need to be corrected (reworded) as follows: Diabetes is often accompanied by disturbances in metabolism, .....
3- The sentence in lines 46-49 needs to be reworded.
4- the sentences in lines 50-52 is not scientifically correct.
5- The overall quality of the English language is satisfactory, but still English language needs to be further improved to meet the required quality of publication.
Author Response
please see the attachment, thank you!

Round 2
Reviewer 2 Report
The authors responded to the issues raised about the quality of the manuscript, however, some of the responses are insufficient. The following points need to be considered:
1- The authors may need to highlight that lyophilization resulted in aggregation, and that lyophilization procedure may need to be optimized.
2- The authors may need to highlight that another more accurate analytical method like LC/MS may need to be used to confirm integrity of insulin molecule and structure. They may also need to cite some literature to support the use of RP-HPLC to test insulin integrity.
The manuscript can be eligible for publication after these issues are considered, and sufficient text is added.
English language is satisfactory.
